# Incidence, Prevalence, and Sources of COVID-19 Infection among Healthcare Workers in Hospitals in Malaysia

**DOI:** 10.3390/ijerph191912485

**Published:** 2022-09-30

**Authors:** Abdul Aziz Harith, Mohd Hafiz Ab Gani, Robin Griffiths, Azlihanis Abdul Hadi, Nor Aishah Abu Bakar, Julia Myers, Maznieda Mahjom, Rosnawati Muhamad Robat, Muhammad Zulfakhar Zubir

**Affiliations:** 1Occupational and Aviation Medicine Department, University of Otago Wellington, New Zealand, 23A Mein Street, Newtown, Wellington 6242, New Zealand; 2Occupational Health Research Centre, Institute for Public Health, National Institutes of Health, Ministry of Health, Shah Alam 40170, Malaysia; 3Medical Development Division, Ministry of Health Malaysia, Aras 3-7, Blok E1, Pusat Pentadbiran Kerajaan Persekutuan, Wilayah Persekutuan Putrajaya 62590, Malaysia; 4Occupational Health Unit, Institute for Medical Research, National Institutes of Health, Ministry of Health Malaysia, Shah Alam 40170, Malaysia

**Keywords:** prevalence, infection rates, incidence rates, COVID-19, healthcare workers, health workers, hospital, source, mortality

## Abstract

The COVID-19 pandemic introduced significant novel risks for healthcare workers and healthcare services. This study aimed to determine the prevalence, trends, characteristics, and sources of COVID-19 infection among healthcare workers during the early COVID-19 pandemic in Malaysian hospitals. A cross-sectional study used secondary data collected from a COVID-19 surveillance system for healthcare workers between January and December 2020. Two surges in COVID-19 cases among healthcare workers in Malaysia were epidemiologically correlated to a similarly intense COVID-19 pattern of transmission in the community. The period prevalence of COVID-19 infection and the mortality rate among healthcare workers in Malaysia were 1.03% and 0.0019%, respectively. The majority of infections originated from the workplace (53.3%); a total of 36.3% occurred among staff; a total of 17.0% occurred between patients and staff; and 43.2% originated from the community. Healthcare workers had a 2.9 times higher incidence risk ratio for the acquisition of COVID-19 infection than the general population. Nursing professionals were the most highly infected occupational group (40.5%), followed by medical doctors and specialists (24.1%), and healthcare assistants (9.7%). The top three departments registering COVID-19 infections were the medical department (23.3%), the emergency department (17.7%), and hospital administration and governance (9.1%). Occupational safety and health units need to be vigilant for the early detection of a disease outbreak to prevent the avoidable spread of disease in high-risk settings. The transformation of some tertiary hospitals to dedicated COVID-19 care, the monitoring of new procedures for the management of COVID-19 patients, and appropriate resource allocation are key to successful risk mitigation strategies.

## 1. Introduction

Occupational disease surveillance systems enable occupational and public health physicians to understand existing on-going communicable diseases or non-communicable diseases related to work. In a pandemic, data collection can demonstrate the extent and focus of occupational health, especially early in an outbreak, when the threats and possible risk mitigation measures are still poorly understood. Occupational health personnel need to respond rapidly for such data to be collected in order to establish measures to protect the health of health care workers (HCWs). One of the earliest studies among HCWs by Nguyen et al. found that HCWs initially had an almost 12 times higher risk of getting COVID-19 than the general population in the United Kingdom (UK) and the United States (US) [1]. Reviewing the 2020 pandemic occupational health surveillance data can enable occupational health services to establish prompt and effective responses early in future pandemics during periods before vaccination, and rapid antigen testing (RAT) surveillance can mitigate risks [2].

Generally, HCWs are at a higher risk of contracting COVID-19 infection when compared to the general population because of their frequent exposure to COVID-19 patients in healthcare facilities and hospitals [3,4]. The intensity of COVID-19 infection within the community served by such health services also poses occupational and non-occupational infection risks for front-line and outward-facing healthcare staff [5,6]. Based on studies conducted in Hong Kong, Japan, Singapore, Taiwan, Thailand, and Vietnam by Lan et al. in 2020 on COVID-19 infection among all essential workers, healthcare workers (HCWs) were the most affected, followed by drivers and transport workers, services and sale workers, cleaning and domestic workers, and public safety workers [5].

Worldwide, the period prevalence of COVID-19 infection among health workers in 2020 was estimated by the World Health Organization (WHO) to be approximately 14% (one in seven healthcare workers), although in some countries, it was as high as 35% [7]. An Omani study found that the prevalence of SARS-CoV-2 among HCWs in tertiary hospitals was 21.2% [8], while in a New York tertiary hospital, it was 9.8% [9]; in a Southern Italian hospital, it was 3.4% [10]; and the lowest prevalence was found in a hospital in a Saudi Arabian hospital, recorded as 0.0% [11]. The burden of COVID-19 on health workers was found to be the lowest among countries in Asia, probably due to their prior experience with SARS-1 [12]. The preparation and responsiveness of the preventive measures in these countries most likely contributed to these better outcomes [13,14]. Additionally, the lower burden also reflected the availability and effectiveness of Occupational Safety and Health Management Systems (OSHMS) in healthcare facilities [12].

The risk of being infected was also dependent on the critical phase of the pandemic, as patients with COVID-19 were not the only source of infection. HCWs were also exposed to infected colleagues or family members who lived in communities of active transmission [6] or were exposed to infected contacts during crowded events, such as wedding receptions and religious gatherings [15]. A study by Jameela Alajmi et al., showed that 95% of COVID-19 infections in health workers were from occupational exposures in non-COVID-19 facilities, while only 5% were from COVID-19 designated facilities [14]. Likewise, 45% of acquired infections were through inadvertent exposure to an infected colleague, and only 29% were from patients [16]. A study in Oman also suggested that the majority of HCW infections (61.3%) were community-acquired [8].

In the early stages of the pandemic, the most prominent factors leading to COVID-19 infections among HCWs were the inadequacy of personal protective equipment (PPE), the increased likelihood of exposure to large numbers of infected patients or long working shifts, inadequate training for infection prevention and control, poor infection control measures, and exposure to unrecognized COVID-19 patients [8,10,17,18]. A few studies found that other factors may or may not also contribute to COVID-19 infection, such as gender, ethnicity, comorbidities, and work overload [1,19,20]. The purpose of this study was to determine the prevalence, characteristics, and sources of COVID-19 infection among health workers in Malaysia. Understanding the extent of the threat was a crucial aspect of developing measures to protect HCWs and maintain a functioning and sustainable healthcare system.

## 2. Methods

### 2.1. Study Subjects

This study was conducted among 155,578 healthcare workers in 144 government hospitals and 3 institutes of the Ministry of Health (MOH) throughout Malaysia. Data were collected during the first year of the COVID-19 pandemic between Jan 2020 and December 2020. The study utilised secondary data collection from a COVID-19 surveillance system for HCWs, including medical doctors (house officers, medical officers, and specialists), medical assistants, nurses (staff nurse, community nurse), pharmacists and assistants, medical laboratory personnel, healthcare assistants, dentists and assistants, administrative personnel, hospital drivers, food preparation personnel, radiographers, occupational therapists and physiotherapists, environmental health professionals, dieticians and nutritionists, support services personnel (security personnel, general administrative assistants, welfare personnel, community assistant personnel), and other clinical personnel.

### 2.2. Screening and Testing of Healthcare Workers

During the first year of COVID-19 in Malaysia, the screening of healthcare workers was carried out in selected target groups based on exposure risk, medium-risk and high-risk exposure; asymptomatic healthcare workers with household contact who were investigated as a person under investigation (PUI) for COVID-19; and symptomatic healthcare workers. This protocol followed the Management of Healthcare Workers (HCW) During COVID-19 Outbreak Guidelines (Annex 21) of the Ministry of Health of Malaysia during the early pandemic in 2020 [21].

### 2.3. Standard Operating Procedures for COVID-19 Testing

In 2020, all suspected COVID-19-positive cases in Malaysia were diagnosed using the SARS-CoV-2 real-time polymerase chain reaction (RT-PCR) test. The MOH produced guidelines for Standard Operating Procedures (SOPs) for laboratory testing [22]. The guidelines were in line with the World Health Organization’s Interim Guidance on laboratory testing for coronavirus disease (COVID-19) in suspected human cases [23].

### 2.4. Investigation of COVID-19 Infection

All COVID-19 cases involving healthcare workers must be notified to the healthcare authorities in accordance with the Occupational Safety and Health (Notification of Accident, Dangerous Occurrence, Occupational Poisoning, and Occupational Disease) Regulations. Health inspectors are in charge of investigating the episode to determine the epidemiological link and any potential source of infection within healthcare facilities. Before it can be endorsed as work related or not work related, an occupational health physician reviews the investigation results. Pending investigation during data collection, the case is marked as undetermined.

### 2.5. Data Sources

At the end of March 2021, data on infected healthcare workers were extracted from the COVID-19 surveillance system under the aegis of the Occupational, Safety, and Health (OSH) Unit, Medical Development Division, MOH, Malaysia. The collected data incorporated the prevalence of COVID-19 in Malaysia and its states (13 states and 2 federal territories), demographic characteristics (age, gender, race, comorbidity, state), occupational details (hospital department, hospital categories, occupational categories), and COVID-19 status (possible source of infection).

### 2.6. Data Analysis

All the data were cleaned and checked for data correctness, data consistency, and usability by identifying error or corruptions in the data, correcting or removing incorrect or duplicate data, and manually processing incomplete data to clear errors. Data were analysed using Statistical Package for the Social Sciences (SPSS) version 28 Mac (IBM SPSS Inc.). An epidemiological week was used to simplify a standardised method of counting weeks to allow data to be compared within a year. The incidence rate describes disease occurrence in a population as a measure of the number of new cases (“incidence”) per unit of time (“rate”). The prevalence rate, on the other hand, is the proportion of people in a population who have a specific disease over a given time period.

## 3. Results

There were two significant surges in COVID-19 infections among healthcare workers in Malaysia during the year 2020, as shown in Figure 1. The first surge of cases occurred during the epidemiology weeks 11 to 17. It started two weeks after the second COVID-19 community epidemic wave in Malaysia (epidemiology week 9). The second surge of COVID-19 cases was registered mainly in Sabah, trending upwards 3 weeks before the announcement of the third COVID-19 epidemic wave (epidemiology week 41). The propagation of COVID-19 infection in Sabah during the third epidemic wave was similarly seen in Selangor, Penang, WPKL, Johor, and Perak (epidemiology weeks 41 to 53).

The COVID-19 incidence rates among healthcare workers correlated to the COVID-19 community incidence rate in the general population, as shown in Figure 2. However, it was noted that throughout most of the epidemiology weeks, the incidence rates per 100,000 healthcare workers were much higher than the incidence rates per 100,000 general population members. This was the most evident whenever there was a surge in COVID-19 cases in the general population. The COVID-19 incidence rate also gradually increased towards the end of the year 2020 (epidemiological weeks 35–53). There were also a number of periodic cycles, where the incidence rates among healthcare workers were more than double the incidence rates among the population. Based on the COVID-19 incidence risk ratio, healthcare workers had a 2.9 times higher risk of getting COVID-19 infection than the general population.

As shown in Table 1, there were 1608 COVID-19 cases among healthcare workers in 2020, implying that the COVID-19 period prevalence among healthcare workers in Malaysia was 1.03%. A further breakdown by states showed that the highest prevalence was in Sabah (3.7%), followed by Selangor (2.11%), WP Labuan (1.09%), WPKL and Putrajaya (0.77%), Negeri Sembilan (0.74%) and Sarawak (0.67%). In contrast, Terengganu recorded the lowest prevalence (0.03%) among the states of Malaysia, followed by Perlis (0.04%) and Pahang (0.05%). Furthermore, there were only three deaths among Malaysian healthcare workers in 2020, with a COVID-19 mortality rate among healthcare workers of 1.9 per 100,000 healthcare workers.

Demographically, the majority of healthcare workers infected with the COVID-19 virus were 24–44 years old, 24–35 years old (58.7%), and 35–44 years old (23.5%). The majority were female (67.7%), Malay (49.7%), and Sabahan (37.6%), with only 8% having comorbidities. Occupationally, the highest numbers of COVID-19 infections were registered in the medical department (23.3%), followed by the emergency department (17.7%), hospital administration and governance (9.1%), anaesthetic department (8.6%), and clinical support services (7.9%). A further detailed categorization by hospital types showed that more than one-third of COVID-19 cases among healthcare workers were contracted in state hospitals (34.8%), and almost one-third of cases were acquired in the major hospitals (31.2%), while one-fifth (19.5%) of the COVID-19 cases were recorded in non-specialist hospitals (Table 2).

A large proportion of COVID-19 infections among healthcare workers were concentrated within the nursing profession occupational group (40.5%), followed by medical doctors and specialists (24.1%), healthcare assistants (9.7%), medical doctor assistants (9.1%), and administrative personnel (3.4%). Eight occupational categories had infection rates higher than the overall average infection rates among healthcare workers (1.03). The highest infection rates per 100 healthcare workers were found among the environmental health professionals (3.33), medical doctor assistants (1.77), food preparation personnel (1.41), medical doctors and specialists (1.26), hospital drivers (1.13), dentists and assistants (1.08), healthcare assistants (1.08), and nursing professionals (1.06). The lowest infection rates were observed among dieticians and nutritionists (0.28), support service personnel (0.32), occupational therapists and physiotherapists (0.50), administrative personnel (0.62), and other clinical personnel (0.66) (Table 3).

The investigation by healthcare authorities (occupational and environmental health professionals) in hospitals found that the majority of infections were acquired in the workplace (53.3%); a total of 36.3% occurred among staff; and 17.0% occurred between patients and staff, as shown in Table 4. However, a large number of COVID-19 infections were community-acquired, at 43.2%, with a minimal balance of 3.5% being unidentifiable. For most occupational groups, the majority of transmissions occurred within their workplace, except for dentists and assistants (85.8%), medical imaging technicians (85.0%), administrative personnel (57.4%), pharmacists and assistants (52.8%), and hospital drivers (51.7%), whose source of infection was more often the community.

## 4. Discussion

Generally, the 1.03% period prevalence of COVID-19 among Malaysian HCWs in 2020 was low compared with the global HCW prevalence estimates during the early pandemic. A report based on pooled data indicated that the 2020 prevalence of COVID-19 among HCWs might have been as high as 51.7% [25], while another study on the global burden of the COVID-19 pandemic among HCWs found that the median prevalence was 10.04% [12]. More recently, the prevalence of COVID-19 among HCWs did appear to be trending lower, with one study reporting a 2021 prevalence of only 11% based on pooled data [26].

In this study, Selangor, Sabah, and Labuan were the three states with high prevalence, surpassing Malaysia’s national prevalence record. Although the Malaysian HCW occupational COVID-19 infection prevalence was low, it still had the potential to significantly impact the delivery of healthcare services when there was a shortage of healthcare workers due to them being infected or more generally affected by the pandemic. Testing targeted and selected occupational groups could also reduce the likelihood of underestimating infections. This was demonstrated by the high incidence of asymptomatic COVID-19 infections recorded when the mass screening of HCWs was undertaken in the Northern Emirates [27]. However, a mass-screening approach seems unsustainable in the long term, as it involves the commitment of considerable human and financial resources.

The Selangor State Health Department and Kuala Lumpur Federal Territory Health Department requested additional healthcare workforce mobilization during the 2nd wave of the COVID-19 community epidemic [28]. This happened due to a “super spreader event” from a mass religious gathering, which assembled approximately 16,000 participants (0.05% out of the 32 million Malaysia population), of which 14,500 were Malaysians and 1500 were from overseas [29]. A comparable scenario was observed in Sabah during the third wave of the community epidemic, when the infection rate among healthcare workers increased rapidly due to the Sabah general election [30], as shown in Figure 1. The propagation of COVID-19 cases from Sabah to Selangor after the election can be seen in Figure 1, when all Malaysian political parties and Sabahans returned to Peninsular Malaysia, resulting in the spread of COVID-19 throughout the Selangor community, where they primarily reside. The MOH mobilized more than 500 HCWs to Sabah (epidemiology week 41) when infection rates among healthcare workers were uncontrolled to overcome the shortfall of healthcare workers in Sabah [31].

As observed in Sabah, it appeared that there were instances where the infection rates among healthcare workers were three times higher than the community infection rates; this had the potential to destabilize healthcare services. A similar scenario happened in Selangor’s hospitals during epidemiology weeks 51–53, when extra workforces were supplemented by transfers or mobilization from other departments within the same hospital [32,33]. In these cases, it is critical for hospital management to nominate HCWs who are at the lowest infection risk to be mobilised, such as those who are younger, have no comorbidities, and are not pregnant. In addition, four hospitals in Selangor were converted to dedicated COVID-19 hospitals, and other medical department services were suspended, except for emergency cases. Medical doctors, including specialists from different departments, were pooled to manage the COVID-19 surge. Hospital transformation to COVID-19 hospitals started with Sungai Buloh Hospital on 2 October 2020, followed by Ampang Hospital, Selayang Hospital, and Serdang Hospital [34]. The action was necessary due to an increase in COVID-19 cases in the State. More stringent health and safety Standard Operating Procedures (SOPs) were more consistently enforced in dedicated COVID-19 hospitals, and disciplinary action was taken when infected healthcare workers were identified as not having complied with these SOPs [16,35].

Demographically, over sixty to seventy percent of COVID-19 infections among HCWs were observed in those who were aged less than 45 years old [8,36]. This could have been due to the fact that the older group worked in specialized areas or served in low-patient-contact roles, though other research found that the trend in infection rates skewed more towards the older age group [1]. Around the world, nurses, a high proportion of whom are female, were the professional group most frequently infected with COVID-19, consistently with the findings of this study [9,37,38]. COVID-19 infection among healthcare workers in this study was skewed to the Sabahan Indigenous group, triggered by the sudden surge of cases in Sabah due to the general election.

Considering health aspects, less than ten per cent of infected healthcare workers in this study had at least a single comorbidity. This could explain the relatively low death rates among healthcare workers in Malaysia as compared with India, which reported HCWs with higher comorbidities with consequently high mortality [39]. Advances in healthcare facility standards and improvements in technology might explain why the Qatar hospital study showed no deaths, although it had similar infection prevalence but a lower mortality rate than India [16].

Emergency departments and medical departments are recognized as established high-risk areas for contracting COVID-19 infection. The similarity between these two departments is that they directly interact with and treat COVID-19 patients. Additionally, emergency department staffs are usually in contact with patients of unknown COVID-19 infectivity status.

Minimum and low compliance with PPE precautions may occur during the early stages of a pandemic, as was seen with the COVID-19 pandemic [40], while the adequacy of PPE could be another issue [1,6]. The OSH team are able to alert and advise high-risk departments on potential hazards and appropriate hazard control specific to the department. Measures might include the application of appropriate PPE, mask fit testing, social distancing, the optimization of airborne engineering control, an increased frequency of disinfection and sanitation, and daily health risk assessment [16]. A study in Sabah revealed that factors associated with COVID-19 transmission among healthcare workers in Sabah included: staying with an infected person, direct physical contact with infected colleagues without PPE, contact with a contaminated COVID-19 environment without wearing appropriate PPE, not adhering to physical distancing with a patient of unknown COVID-19 infection status, and lack of training in prevention and infection control [40]. In contrast, in Saudi Arabia, zero cases were recorded in an anaesthetic department where the highest standards of infection control were applied [11], even though tracheal intubation is seen as the riskiest procedure in anaesthesia [41].

Nurses were found to be the most likely to be infected with the COVID-19 virus in international studies in HCWs. Similar in Malaysia, but the infection rates among nurses were not the highest compared with other healthcare professions. In this study, despite being the most vulnerable group in terms of likely exposure to sick patients, infection rates among nurses were only in the eighth position behind environmental health professionals, medical doctor assistants, food preparation workers, medical doctors and specialists, personnel hospital drivers, healthcare assistants, and dentists and assistants.

High infection rates among specific occupational categories could interrupt an organisation’s capacity to function and carry out specific activities associated with specific occupational categories. However, looking at the source of infection, less prominent occupational categories, such as radiographers, dentists and assistants, pharmacists and assistants, dieticians and nutritionists, and environmental health professionals, were more likely to get infected by the community rather than in a work-related environment. The sources of COVID-19 infection among healthcare workers should be carefully analysed, as almost fifty percent of the infections were sourced from the workplace and the rest were from the community.

Based on our data, most transmissions among staff occurred at mealtimes, meetings, nurse stations, and in changing rooms. These transmissions could be prevented if strict preventive measures and outbreak prevention protocols are applied. Similar to a study in Oman, in this study, the source of infection for a small portion of the infected healthcare workers could not be determined [8]. Although undetermined percentages contribute only 3.5%, it should be noted that these data were collected at the end of March 2021, three months after the year 2020 ended. Due to the pandemic’s high number of COVID-19 cases in the community and healthcare workers, the majority of cases were pending investigation by Sabah healthcare authorities when the analysis was undertaken.

## 5. Conclusions

Early detection and isolation are critical, especially for nursing professionals, medical doctors and specialists, and healthcare assistants who are in close contact with patients and are likely to be infected. Healthcare organisations must plan ahead of time to ensure patient safety and the safety of the entire health workforce. The COVID-19 pandemic has thrown into sharp relief that the hazards of exposure to infection in health care settings both pose an unprecedented occupational health risk for healthcare workers and also threatens the sustainability of critical care services in a public health crisis. The most important strategy for sustaining healthcare services and the workforce is to reallocate resources and prioritise managing unprecedented COVID-19 risks to healthcare workers via prompt and evidence-based occupational and public health responses. The transformation of tertiary hospitals into COVID-19 hospitals, the monitoring of new procedures for the management of COVID-19 patients, and appropriate resource allocation are key to successful risk mitigation strategies.

## 6. Limitations and Future Research

Despite the fact that this study relied on secondary data, the researchers attempted to minimise the study’s limitations. There are three limitations to this study. There was a lack of reliable data among the secondary data collected for this study, primarily due to missing data. The researchers attempted to supplement the missing data by contacting ground sources in relevant hospitals. The second limitation was data availability. This study’s research questions were restricted to the availability of specific information from existing secondary data and could not be extended to researchers’ other research interests. Finally, because exposure and outcomes were assessed concurrently, this study was unable to generate a temporal relationship between exposure and outcomes. However, this study could inform the exact burden of COVID-19 among HCWs during the first year of the pandemic and identify the gaps and disparities in COVID-19 preventive practice among healthcare workers at the national level.

Future research should focus on interventional studies that could prevent COVID-19 transmission among healthcare workers in healthcare facilities. A qualitative research design would be extremely beneficial in issues concerning healthcare workers in hospitals during the COVID-19 pandemic. This study design could be extremely useful for occupational safety and health professionals for identifying specific issues among healthcare workers and designing appropriate preventive and protective follow-up actions.

## Figures and Tables

**Figure 1 ijerph-19-12485-f001:**
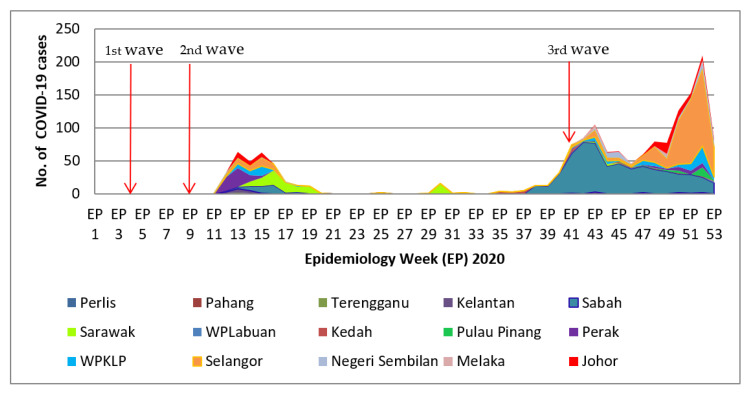
COVID-19 infection among healthcare workers in Malaysian hospitals, 2020 (epidemiology weeks against states). The first wave of community epidemic COVID-19 infection in Malaysia started on 24 January 2020. The second wave started on 27 February 2020, and the third wave was announced on 8 October 2020.

**Figure 2 ijerph-19-12485-f002:**
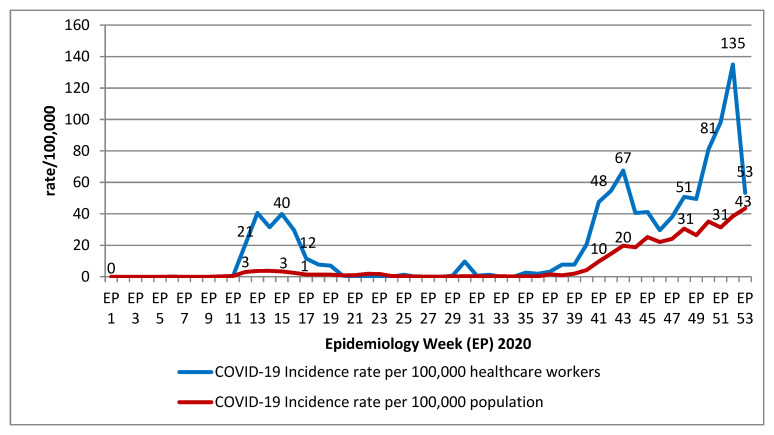
COVID-19 incidence rates among healthcare workers (per 100,000 healthcare workers) and the general population (per 100,000 populations). Data for COVID-19 infection in the population were obtained from open access COVID-19 data from the website of Ministry of Health of Malaysia [24]. The cumulative COVID-19 incidence rates among healthcare workers and population were 1033 per 100,000 healthcare workers and 352 per 100,000 populations, respectively.

**Table 1 ijerph-19-12485-t001:** Prevalence and mortality rate of COVID-19 infection among healthcare workers in government hospitals in Malaysia, 2020.

Variables	Total Staff ^a^	n	%
Malaysia	155,578	1608	1.03
**States**			
Perlis	2145	1	0.04
Kedah	10,334	25	0.24
Pulau Pinang	8861	28	0.31
Perak	15,052	90	0.59
Selangor ^b^	20,920	442	2.11
WPKL and Putrajaya ^c^	13,997	108	0.77
Negeri Sembilan	6696	50	0.74
Melaka	4987	19	0.38
Johor	15,887	76	0.47
Pahang	9419	5	0.05
Terengganu	6487	2	0.03
Kelantan	7604	20	0.26
Sabah	17,271	640	3.70
Sarawak	14,240	96	0.67
WP Labuan	550	6	1.09
**Mortality rate**			
Death	155,578	3	0.000019

^a^ Source: Medical Development Division, Ministry of Health; ^b^ Includes National Cancer Institute; ^c^ Includes Kuala Lumpur Hospital and National Blood Bank.

**Table 2 ijerph-19-12485-t002:** Personal and occupational characteristics of healthcare workers infected with COVID-19 in Malaysian hospitals (N = 1608).

Personal Characteristics	n	%
**Age Group**		
≤24 years old	65	4.0
25–34 years old	944	58.7
35–44 years old	378	23.5
45–54 years old	165	10.3
≥55 years old	32	2.0
**Gender**		
Male	520	32.3
Female	1088	67.7
**Race**		
Malay	799	49.7
Chinese	74	4.6
Indian	92	5.7
Sabah Indigenous	604	37.6
Sarawak Indigenous	30	1.9
Others	9	0.5
**Comorbidity**		
Single comorbidity	93	5.8
Double comorbidities	29	1.8
Triple comorbidities	4	0.2
>3 comorbidities	2	0.1
No comorbidities	1480	92.0
**Occupational Characteristics**		
**Hospital Department**		
Medical	374	23.3
Emergency	284	17.7
Hospital admin and governance	147	9.1
Anaesthetic	138	8.6
Hospital clinical support services	128	7.9
Surgery	118	7.3
Pathology	71	4.4
Obstetrics and gynaecology	63	3.9
Pharmacy	55	3.4
Paediatric	52	3.2
Ward (multidiscipline)	43	2.7
Orthopaedics	37	2.3
Radiology	33	2.1
Dentistry	6	0.4
Unidentified	59	3.7
**Hospital Category**		
State hospital	560	34.8
Major hospital	502	31.2
Minor hospital	123	7.6
Non-specialist hospital	313	19.5
Institute hospital	110	6.8

**Table 3 ijerph-19-12485-t003:** Infection rates and distribution of healthcare workers infected with COVID-19 in Malaysian hospitals (N = 1608).

Occupational Category	Total Staff ^a^	n (%)	Infection Rate (%)
Environmental health professionals	90	3 (0.2)	3.33
Medical doctor assistants	8268	146 (9.1)	1.77
Food preparation personnel	1991	28 (1.7)	1.41
Medical doctor and specialists	30,830	387 (24.1)	1.26
Hospital drivers	2565	29 (2.3)	1.13
Dentists and assistants	555	6 (0.4)	1.08
Healthcare assistants	19,232	156 (9.7)	1.08
Nursing professionals	61,480	652 (40.5)	1.06
Radiographer	2324	20 (1.2)	0.86
Pharmacists and assistants	6248	49 (3.0)	0.78
Laboratory personnel	4951	37 (2.3)	0.75
Other Clinical personnel	602	4 (0.2)	0.66
Administrative personnel	12,031	74 (4.6)	0.62
Occupational and physiotherapists	1981	10 (0.6)	0.50
Support services personnel	1876	6 (0.4)	0.32
Dieticians and nutritionists	352	1 (0.1)	0.28
Total (average infection rate)	155,376	1608	1.03

^a^ Source: Medical Development Division, Ministry of Health.

**Table 4 ijerph-19-12485-t004:** Source of COVID-19 infection among healthcare workers in Malaysian hospitals in 2020.

Occupational Category	n	Source of Infection			
		Work, n (%)		Community, n (%)	Undetermined, n (%)
		Staff to Staff	Patient to Staff		
Nursing professionals	652	237 (36.3)	116 (17.8)	276 (42.3)	23 (3.5)
Medical doctor and specialists	387	137 (35.4)	101 (26.1)	135 (34.9)	14 (3.6)
Healthcare assistants	156	55 (35.3)	20 (12.8)	73 (46.8)	8 (5.1)
Medical doctor assistants	146	55 (37.7)	30 (20.5)	58 (39.7)	3 (2.1)
Administrative personnel	72	26 (36.1)	0 (0.0)	44 (61.1)	2 (2.8)
Pharmacists and assistants	49	19 (38.8)	4 (8.2)	26 (53.1)	0 (0.0)
Laboratory personnel	37	17 (45.9)	0 (0.0)	18 (48.6)	2 (5.4)
Hospital drivers	29	13 (44.8)	1 (3.4)	15 (51.7)	0 (0.0)
Food preparation personnel	28	16 (57.1)	0 (0.0)	11 (39.3)	1 (3.6)
Radiographer	20	1 (5.0)	0 (0.0)	17 (2.4)	2 (10.0)
Occupational and physiotherapists	10	3 (30.0)	0 (0.0)	7 (85.0)	0 (0.0)
Support services personnel	7	3 (42.9)	0 (0.0)	4 (57.1)	0 (0.0)
Dentists and assistants	7	0 (0.0)	0 (0.0)	6 (85.7)	1 (14.3)
Other clinical personnel	4	2 (50.0)	1 (25.0)	1 (25.0)	0 (0.0)
Environmental health professionals	3	0 (0.0)	0 (0.0)	3 (100)	0 (0.0)
Total	1608 (100)	584 (36.3)	273 (17.0)	695 (43.2)	56 (3.5)

## Data Availability

Restrictions apply to the availability of these data. Data were obtained from Occupational Safety and Health Unit, Medical Development Division, Ministry of Health Malaysia and are available from the corresponding author with the permission of Director of Medical Development Division, Ministry of Health Malaysia.

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
