# Peer review of "Incidence, Prevalence, and Sources of COVID-19 Infection among Healthcare Workers in Hospitals in Malaysia"

_ijerph, 2022, doi:10.3390/ijerph191912485_

Round 1
Reviewer 1 Report
I would like to thank the authors for this research that aims to determine the prevalence, characteristics, and sources of COVID-19 infection among health workers in Malaysia. Understanding the extent of the threat was a crucial aspect of developing measures to protect health workers and ensure a functioning and sustainable healthcare system.
The research subject is timely, innovative, and highly interesting. It also fits the aim and scope of the journal.
The research is well designed and follows a sound scientific research method.
Results are clear and could have an impact among the community of researchers.
However, the research needs several adjustments:
The manuscript still needs an intensive language editing as it contains several errors as highlighted in the attached document.
Lines 161-163 you said: “There were also a number of periodic cycles, where the incidence rates among healthcare workers were more than double the incidence rate among the population”. Is there any logic explanation of the very high incidence rate between episodes 49 to 53?
Table 1: Are there a major issue in Sabah and Selangor districts comparing to Kedah or Perak. Is the difference in the prevalence associated with good or bad practices / bad or better preventives measures, trainings … associated with the health units?
You need to add the recommendations, limitations and implications of your research.
You need also to explain us the importance of the learned lessons from COVID-19 especially in a period in which we do not talk any more about this pandemic.
You can solidify your finding by recent references.
Other minor comments are directly attached to the manuscript.

Author Response
Response to Reviewer Comments
Point 1: Grammatical Changes.
- Reviewer suggest making changes in word use; Source to Sources.
Response 1: The authors follow reviewers' suggestions
Point 2: To support and solidify the statements
- 5 more citations in the introduction.
- 1 more citation at the discussion part.
Response 2: The authors follow reviewers' suggestions. We have made changes according to the references given.
Point 3: Check sentence
- Line 124
- Line 262
- Line 325
Response 3: Authors made correction as suggested.
- Line 124 :
- Line 262 :
- Line 325
Point 4: Are there any logic explanation of the very high incidence rate between episodes 49 to 53 based on figure 2?
Response 4: Malaysia is divided into two parts: Malaysia Peninsular and Malaysia East, separated by the South China Sea. Figure 2 shows that cases of COVID-19 infection among healthcare workers are gradually increasing following the third wave of the COVID-19 epidemic in Malaysia, as shown in Figure 1, which began in Sabah during the general election. Meanwhile, all Malaysian political parties and Sabahans from the peninsular travelled to Sabah for the campaign. As a result, following the election, members of political parties return to Peninsular Malaysia. They primarily resided in Selangor province, which has a high population density, resulting in the spread of COVID-19 throughout the Selangor community.
Point 5: Are there a major issue in Sabah and Selangor districts comparing to Kedah or Perak. Is the difference in the prevalence associated with good or bad practices / bad or better preventives measures, trainings … associated with the health units?
Response 5: The spillover effects is the main answer on his as tremandous increasing fo COVID-19 cases in community led to increase cases among healthcare workers. Further explanation has been included in the discussion at line 246 – 252.
“ The propagation of COVID-19 cases from Sabah to Selangor after the election can be seen in Figure 1, when all Malaysian political parties and Sabahans return to Peninsular Malaysia, resulting in the spread of COVID-19 throughout the Selangor community, where they primarily reside.”
Point 6: You need to add the limitations and implications of your research
Response 6: The authors followed the reviewers' suggestions and added the limitation and implication components as suggested.

Reviewer 2 Report
The authors describe the occurrence of COVID-19 among health care workers in hospitals in Malaysia in 2020. They have data from almost 156,000 health care workers from about 150 hospitals. This data is also compared to the incidence of COVID-19 in the population and the source of the infection is examined. Overall, health care workers had a 2.9 times higher risk of developing COVID-19 than someone in the general population.
Major comments
1. The authors state that there were three waves of COVID-19 in the general population. Nevertheless, when looking at Figure 2, I only recognize ‘wave 3’ in the incidence rate of the general population.
2. Results, lines 184-194. In this part, the authors give a description of the characteristics of the cases. However, no information of the total group of health care workers in the analyses is given. In lines 197-208, this is done for the different occupational categories, but what about the age and sex distribution of the total group? And is comorbidity also available for the non-cases? These data give more insight in the persons with the highest risk of developing COVID-19. This could also be used for an elaboration on this theme in the discussion in lines 267-281.
3. Source of the infection is also reported (i.e. lines 212-220, Table 4). How was this established? Maybe add a short description of this in the Methods? Furthermore, for the occupation categories Nursing professionals, Medical doctors & specialists, and Health care assistants, the source often remained undetermined. This is probably due to the fact that these are categories working the most and the closest with patients. I would suggest to elaborate on these results some more in for example the discussion.
4. Discussion, lines 303-318. Here the overall infection rates are compared and discussed between the occupational categories. It is stated that nurses do not have a higher infection rate than for example personnel hospital drivers. However, the first group appeared to be especially infected during their profession (or undetermined) whereas the latter group was mostly infected in the general population. I would suggest to particularize here a bit more.
Minor comments
5. Throughout the manuscript, abbreviations are used, but only used once (for example UK, US) or also explained several times (for example SOP, OSH) or after already have used the abbreviation several times (for example PPE). Please check the text on consistent and efficient use of abbreviations.
6. Introduction, lines 46-48. In the beginning of the sentence and at the end the word ‘early’ is used. I would suggest to remove the first ‘early’.
7. Introduction, lines 67-70. I don’t understand this sentence. Do the authors mean that around 1 in 7 cases is a health care worker? Or that 1 in 7 health care workers had had COVID-19 in 2020?
8. In the results the term ‘epidemiology week’ is introduced. I assume it is the number of the week in the year, but please specify it (short) in the Methods.
9. Results, lines 177-179. Maybe add ‘of COVID-19’ to ‘mortality rate’ as it is not the overall mortality rate in health care workers but rather the fatality rate due to COVID-19.
Author Response
Response to Reviewer Comments
Point 1: The authors state that there were three waves of COVID-19 in the general population. Nevertheless, when looking at Figure 2, I only recognize ‘wave 3’ in the incidence rate of the general population.
Response 1: The epidemiological COVID-19 first wave was flattened due to a low number of COVID-19 cases and a massive increase in COVID-19 cases in the subsequent wave. That is why the graft was aided by the arrow. The epidemiological waves, on the other hand, are determined by Malaysian public health authorities.
Point 2: Results, lines 184-194. In this part, the authors give a description of the characteristics of the cases. However, no information of the total group of health care workers in the analyses is given. In lines 197-208, this is done for the different occupational categories, but what about the age and sex distribution of the total group? And is comorbidity also available for the non-cases? These data give more insight in the persons with the highest risk of developing COVID-19. This could also be used for an elaboration on this theme in the discussion in lines 267-281.
Response 2: The authors will answer based on the reference lines.
- lines 184-194 : The authors have improved the sentences by putting the total number of COVID-19 among healthcare workers in Malaysia.
- lines 197-208 : The total number of sexes and age distribution in the total group were in the sentences and the table.
- Is comorbidity also available for the non-cases? No, as we did not collect data for non-cases.
- Elaboration on this theme in the discussion in lines 267-281 : The authors do agreed with the reviewer. This sentence was added in the discussion “In these cases, it is critical for the hospital management team to choose healthcare workers who are at the lowest risk of being infected to be mobilised, such as those who are younger, have no comorbidities, and are not pregnant”.
Point 3: Source of the infection is also reported (i.e. lines 212-220, Table 4). How was this established? Maybe add a short description of this in the Methods? Furthermore, for the occupation categories of nursing professionals, Medical doctors & specialists, and health care assistants, the source often remained undetermined. This is probably due to the fact that these are categories working the most and the closest with patients. I would suggest to elaborate on these results some more in for example, the discussion.
Response 3: The investigation statement appears in the paragraph before table 4, on lines 226-228. But, the authors agreed with the reviewer to add a short description of "Investigation of COVID-19 infection". At the end of the discussion part, the authors also further discussed the unidentified sources of COVID-19 among healthcare workers.
Point 4: Discussion, lines 303-318. Here, the overall infection rates are compared and discussed between the occupational categories. It is stated that nurses do not have a higher infection rate than, for example, personnel hospital drivers. However, the first group appeared to be especially infected during their profession (or undetermined) whereas the latter group was mostly infected in the general population. I would suggest to particularize here a bit more.
Response 4: The authors took the reviewers' advice and expanded on this theme a little more.
Point 5: Throughout the manuscript, abbreviations are used, but only used once (for example UK, US) or also explained several times (for example SOP, OSH) or after already have used the abbreviation several times (for example PPE). Please check the text on consistent and efficient use of abbreviations.
Response 5: Abbreviations checked
- UK, US
- SOP, OSH
- PPE
Point 6: Introduction, lines 46-48. At the beginning of the sentence and at the end, the word ‘early’ is used. I would suggest to remove the first ‘early’.
Response 6: The authors took the reviewers' advice and delete the first ‘early’.
Point 7: Introduction, lines 67-70. I don’t understand this sentence. Do the authors mean that around 1 in 7 cases is a health care worker? Or that 1 in 7 health care workers had had COVID-19 in 2020?
Response7: 1 in 7 healthcare workers
Point 8: In the results, the term ‘epidemiology week’ is introduced. I assume it is the number of the week in the year, but please specify it (short) in the Methods.
Response 8: The authors took the reviewers' advice and put it in the methodology under data analysis.
Point 9: Results, lines 177-179. Maybe add ‘of COVID-19’ to ‘mortality rate’ as it is not the overall mortality rate in health care workers but rather the fatality rate due to COVID-19.
Response 9: The authors took the reviewers' advice accordingly.

Reviewer 3 Report
This is a very well written study based on secondary data and has very important clinical significance. Here are a couple of comments and can be improved.
1) What are the missing rate for the study population? The authors didn't mention the missingness in covariates at all, looks like a complete dataset. I am a bit suspicious about the potential selection bias. It will be helpful if the authors could describe more on this.
2) What are the detailed vaccine information for the study population? The brand, dose of vaccine. Vaccine could be a confounding factor for the findings they got so far, and could have implications on future occupational advice.
3) It will be helpful the authors could add more suggestions and recommendations in the discussion and conclusion section for health care workers.
Author Response
Response to Reviewer Comments
Point 1: What are the missing rate for the study population? The authors didn't mention the missingness in covariates at all, looks like a complete dataset. I am a bit suspicious about the potential selection bias. It will be helpful if the authors could describe more on this.
Response 1: The authors agreed with the reviewer's concern. The authors made an improvement to the manuscript by adding the limitations of the study. "The researchers, on the other hand, attempt to supplement the missing data by contacting ground sources at relevant hospitals."
Point 2: What are the detailed vaccine information for the study population? The brand, dose of vaccine. Vaccine could be a confounding factor for the findings they got so far, and could have implications on future occupational advice.
Response 2: We did not collect the vaccine information. Anyhow, we have a plan to conduct a study on breakthrough infection among healthcare workers post-complete vaccination.
Point 3: It will be helpful the authors could add more suggestions and recommendations in the discussion and conclusion section for health care workers.
Response 3: Based on the nature of the study design, it is quite difficult for authors to make more suggestions beyond the scope of this study. However, we added a few more discussions and future directions of the study.

Round 2
Reviewer 1 Report
I see that the authors made the necessary changes as suggested.
Author Response
Dear reviewer, thank you for reviewing our manuscript. We had made an improvement on grammars as suggested by the editor. Thank you.
Reviewer 2 Report
None
Author Response
Dear reviewer,
Thank you for reviewing our manuscript. We have made an improvement in the manuscript. The English native speaker has proofread and improved the manuscript.